# The Critical Role of TRIB2 in Cancer and Therapy Resistance

**DOI:** 10.3390/cancers13112701

**Published:** 2021-05-30

**Authors:** Victor Mayoral-Varo, Lucía Jiménez, Wolfgang Link

**Affiliations:** Instituto de Investigaciones Biomédicas “Alberto Sols” (CSIC-UAM), Arturo Duperier 4, 28029 Madrid, Spain; vmayoral@iib.uam.es (V.M.-V.); ljimenez@iib.uam.es (L.J.)

**Keywords:** TRIB2, Tribbles, pseudokinases, signalling, cancer, therapy resistance, biomarker

## Abstract

**Simple Summary:**

The Tribbles proteins are members of CAMK Ser/Thr protein kinase family. They are evolutionary conserved pseudokinases found in most tissues of eukaryotic organisms. This ubiquitously expressed protein family is characterized by containing a catalytically deficient kinase domain which lacks amino acid residues required for the productive interaction with ATP and metal ions. Tribbles proteins exert their biological functions mainly through direct interaction with MAPKK and AKT proteins, therefore regulating important pathways involved in cell proliferation, apoptosis and differentiation. Due to the role of MAPKK and AKT signalling in the context of cancer development, Tribbles proteins have been recently considered as biomarkers of cancer progression. Furthermore, as the atypical pseudokinase domain retains a binding platform for substrates, Tribbles targeting provides an attractive opportunity for drug development.

**Abstract:**

The Tribbles pseudokinases family consists of TRIB1, TRIB2, TRIB3 and STK40 and, although evolutionarily conserved, they have distinctive characteristics. Tribbles members are expressed in a context and cell compartment-dependent manner. For example, TRIB1 and TRIB2 have potent oncogenic activities in vertebrate cells. Since the identification of Tribbles proteins as modulators of multiple signalling pathways, recent studies have linked their expression with several pathologies, including cancer. Tribbles proteins act as protein adaptors involved in the ubiquitin-proteasome degradation system, as they bridge the gap between substrates and E3 ligases. Between TRIB family members, TRIB2 is the most ancestral member of the family. TRIB2 is involved in protein homeostasis regulation of C/EBPα, β-catenin and TCF4. On the other hand, TRIB2 interacts with MAPKK, AKT and NFkB proteins, involved in cell survival, proliferation and immune response. Here, we review the characteristic features of TRIB2 structure and signalling and its role in many cancer subtypes with an emphasis on TRIB2 function in therapy resistance in melanoma, leukemia and glioblastoma. The strong evidence between TRIB2 expression and chemoresistance provides an attractive opportunity for targeting TRIB2.

## 1. Introduction

Tribbles proteins are pseudokinases conserved in metazoan species throughout evolution. Tribbles was first discovered as a regulator of mitosis and morphogenesis in *Drosophila melanogaster* [1]. In human, TRIB 1, 2, 3 and Sgk495/serine/threonine kinase 40 (STK40), are the four members of a subfamily of pseudokinases closest to fly TRIB [2]. The sequence analysis of human TRIB family showed that the pseudokinase domain of TRIB2 and TRIB1 had a high level of homology (71%) while these members had around 53% with TRIB3. However, the homology of STK40 with any of the other family members was less than 22% [2,3]. A taxonomic analysis of the Tribbles sequences revealed TRIB2 as the most ancestral member of the Tribbles protein family as TRIB2 orthologs were identified in the oldest metazoans, such as cnidarians and sponges while orthologs of TRIB1 and 3 were detected in later metazoans linages that led to vertebrates [4]. This protein family has been classified as serine/threonine pseudokinases without ATP affinity (TRIB1) or low affinity (TRIB2 and TRIB3) and with no or only residual phosphotransferase capacity in vitro [4].

Pseudokinases have been less studied than canonical kinases although about 10 percent of kinases encoded in the human genome are pseudokinases [5]. Pseudokinases are characterized by the lack of at least one of the conserved amino acids essential for the binding of ATP and/or catalysis and are classified in several subfamilies including the Tyr pseudokinases erythropoietin-producing hepatocellular carcinoma (Eph) A10 and EphB6, Tribbles/TRIB/ STK40 and the orphan pseudokinase protein serine kinase histone (PSKH) 2, whose biological function still remains unknown despite clear conservation in most vertebrates [2]. The Tribbles/TRIB/STK40 family has been classified as serine/threonine pseudokinases whose pseudokinase domain are similar to the active site of canonical kinases CAMK (calcium/calmodulin-dependent protein kinase) [2,6].

The structure of the human TRIB protein can be divided in three parts: the N-terminal PEST region (70–100 residues), atypical bilobed central pseudokinase domain, and a C-terminal tail (approximately 25 residues) that binds E3 ligases [7]. The N-terminal domain is characteristic of each member of the TRIB family with specific motifs/sequences [8,9] which are essentials for the stability and regulation of the proteins [4,10,11]. The pseudokinase domain is highly conserved during evolution and is homologous to serine/threonine kinase domains. The most striking difference between conventional kinases and the pseudokinase domain of Tribbles proteins is the absence of an Asp–Phe–Gly (DFG) tripeptide motif which chelates the Mg^2+^ ion regulating ATP binding [11]. As in canonical kinases, the pseudokinase domain is formed by two lobes, the N-lobe and the C-lobe. The N-lobe contains the kinase machinery and the C-lobe the substrate -binding site. The C-terminal tail contains the HPW (F/L) and DQXVP (D/E) sequences. While the HPW (F/L) sequence is involved in the interactions between MEK1 as well as other MAPKK dual-specificity kinases and TRIB pseudokinases, the DQXVP (D/E) sequence promotes the interaction with ubiquitin E3 ligases such as COP1, β-TRCP, Smurf1 and TRIM21 [4,12]. Moreover, the sequences HPW (F/L) and DQXVP (D/E) have been proposed to interact in cis with the N-lobe and C-lobe of the pseudokinase domain respectively and regulate the interaction with potential substrates [4]. Furthermore, Tribbles proteins contain additional important residues and sequences. The conserved Lys 90 and 177 residues, presented in all Tribbles proteins, have been shown to be required for the interaction of TRIB2 with ATP [6,13], and to be essential for C/EBP-α degradation by TRIB2 [4,14], respectively. C/EBP-α is an important transcription factor and its degradation by TRIB1 or 2 is closely related with leukemia [15]. In addition, a conserved E (S/N) LED sequence has been described that replaces the DFG motif of canonical kinases in Tribbles proteins. The ESLED sequence in TRIB2 has been suggested to play a role in ATP binding and hydrolysis [13].

It is thought that while Tribbles proteins lost their enzymatic activity to phosphorylate substrates, they retained their capability to interact with signalling proteins and function as a scaffold proteins [16]. Tribbles proteins have been shown to play an important role in regulating cell differentiation, proliferation, migration and growth. However, the expression of the three TRIBs and their functional role depends on the tissue and cell type in which they are expressed [17,18]. In hematopoiesis, the expression of TRIB1 is higher in myeloid cells, compared to TRIB2 or TRIB3 that are more expressed in B cells and dormant hemopoietic stem cells, respectively [19]. Furthermore, each member of the Tribbles family has been shown to exert specific physiological functions. While TRIB1 regulates hepatic lipid metabolism [20], TRIB2 regulates pluripotency of embryonic stem cells and enhances their reprogramming efficiency [21], TRIB2 and TRIB3 participate in adipogenesis [22,23] and TRIB3 regulates glucose homeostasis [24]. Though there is accumulated evidence that all members of the TRIB protein family are involved in normal physiology [8,21,23,25,26], in this review we will focus on the specific role of TRIB2 in tumour formation, progression and therapy resistance.

## 2. Unique Structural Features of TRIB2

TRIB2 has a similar structure than the other members of the TRIB protein family but counts with several specific features. The N-terminal domain has been analyzed in different species, showing that in TRIB2 is more evolutionary conserved than in TRIB1 and TRIB3 [11]. As shown in Figure 1, the N-terminal domain contain a sequence rich in serines and prolines, in particular close to the pseudokinase domain, which is characteristic of a PEST sequences known to be involved in the regulation of the half-life of proteins [11]. In the N-terminal domain of TRIB2 the phosphorylation of one of these serine residues, namely serine-83 by p70S6K together with the amino acid sequence 69–85 and the five most N-terminal amino acids, called TRIB2 degradation domain (TDD) are essential for the degradation of TRIB2 via Smurf-1 and β-TRCP in human liver cells [10,27].

The pseudokinase domain is highly conserved among the three Tribbles proteins, differentiating them from the other member of the subfamily, STK40, and the broader family of canonical CAMKs [2]. Indeed, the human TRIB1-3 are unique among all human proteins in combining a kinase-like domain and a domain that promotes the interaction with ubiquitin ligases. The acidic pseudokinase domain of the TRIBs contains an αC-helix and a MEK-binding site at the end of C-lobe and a distinct C-terminal peptide motif that interacts directly with a small cluster of cellular E3 ubiquitin ligases [6]. The C-helix of TRIB2 contains three unusual cysteine residues at the end of the β3-Lys-containing motif which are not present in TRIB1 or 3 leading to a truncated αC-helix in the N-lobe [28]. These residues have been shown to be covalently bound by several kinase inhibitors as discussed below. In addition, residues 158–180 of TRIB2 have been shown to be key for binding NF-κB2 and regulating NF-kB activity in epithelial cells [29]. At the sequence level there are variation between the TRIB family members like E (S/N) LED in the pseudokinase domain HPW (F/L) and DQXVP (D/E) in the C-tail, and in the PEST region, that should be studied in depth to explain the differences in regulation and signaling of each members of the TRIB family [30].

## 3. Regulation of TRIB2 Expression and TRIB2 Signalling

Tribbles proteins seem to have redundant and non-overlapping functions. TRIB2-specific regulation and signaling depends on the cell type and the physiological context. TRIB2 regulation can occur through different means including by transcription factors-mediated activation or repression of TRIB2 expression, through miRNAs or by promoting TRIB2 degradation by the proteasome. Interestingly, the number of predicted transcription factor-binding sites in human and murine TRIB2 gene promoter region is significantly greater compared to TRIB1 or 3 [4]. Table 1 lists transcription factors that have been directly involved in the regulation of TRIB2 in different cell types. Interestingly, it has been described that TRIB2 is expressed in a cyclic manner during the cell cycle, though the underlying mechanism remains to be established [31]. The study of the mechanisms that regulate TRIB2 during the cell cycle might shed light on how its deregulation leads to diseases such as cancer.

The regulation by miRNAs is yet another level of TRIB2 regulation that has emerged recently. The hsa-miR-99b/let-7e/miR-125a cluster downregulates TRIB2 expression and thereby enhances STAT3 activation as a consequence of increased extracellular signal-regulated kinase (ERK) and Jun kinase activation by binding mitogen-activated protein kinase (MAPK) complexes, MKK7 and MEK1 in monocytes [39]. miR-511, miR-1297 and let-7 inhibit proliferation in adenocarcinoma of lung by suppressing TRIB2 expression and consequently increasing C/EBPα expression [40,41]. Similarly, miR-206 and miR-140 suppress lung adenocarcinoma cell proliferation and metastasis via reduced phospho-Smad3/Smad3, which downregulate TRIB2 [34]. On the other hand, miR-505 through inhibition of ZEB1-AS1 upregulates TRIB2 expression in pancreatic cells [42]. In addition, miR-155 induces the expression of TRIB2 as part of a group of apoptosis-associated genes in AML [43].

Moreover, TRIB2 has been shown to be regulated by signalling pathways such as Toll-like receptor 5 (TLR5), Wnt/β-catenin/TCF4 and p70S6K/Smurf1/β-TRCP. Whereas TLR5 ligand stimulation induces TRIB2 expression, TRIB2 mediate a negative feedback by inhibiting TLR5-mediated activation of NF-κB downstream of TRAF6 in both human and mouse colon epithelium [29]. The Wnt/β-catenin/TCF4 pathway promotes TRIB2 expression in liver cancer cells [38], and in this case TRIB2 also produces a negative feedback, via TRIB2-mediated ubiquitination and degradation of TCF4 and β-catenin through the E3 ligase binding region [12]. At the protein level, TRIB2 is regulated by proteasomal degradation via p70S6K/Smurf1/β-TRCP in human liver cells [10,27].

TRIB2 regulates different downstream signaling pathways through its interaction with MAPK, AP4, CDC25, OCT3/4, C/EBP alpha, ubiquitin E3 ligases, PCBP2, MAPK, and AKT. Through these interactions TRIB2 plays a key role in cellular processes like cell cycle, senescence, pluripotency of stem cell, protein degradation, and cell survival. The direct interaction between TRIB2 and MAPK can inhibit the MAPK signalling pathway. TRIB2 has been described to directly interact with MEK1 and MKK7 (MAPKs) through the pseudokinase domain of TRIB2 and the N-terminal region of these MAPKs. However, the C-tail of TRIB2 is also necessary for the interaction with MKK7. While MKK7/TRIB2 has been shown to be localized in both, the cell nucleus and cytoplasm, the MEK1/TRIB2 complex is only localized in the cytoplasm [44]. TRIB2 may also indirectly activate MAPKs, such as p38, which control cell cycle regulators that are aberrantly activated in myeloid leukemia, exerting a tumour suppressor function [45]. In colorectal cancer, it has been described that the pseudokinase domain of TRIB2 interacts with AP4 and enhances AP4-mediated reduction of p21 expression, ultimately leading to decreased senescence and apoptosis of tumour cells [46]. TRIB2 interacts directly with CDC25C via the pseudokinase domain of TRIB2 which leads to the K48-linked polyubiquitination of CDC25C and its subsequent degradation via the proteasome, producing cell cycle arrest [31]. Moreover, TRIB2 is involved in the pluripotency of embryonic stem cells and is important for maintaining self-renewal ability through direct interaction with the transcription factors related with induced pluripotent stem cells, OCT3/4 and increasing OCT3/4 promoter activity [21].

Another important role of the TRIB signalling is the regulation of C/EBP-α. The three TRIBs may interact with C/EBP-α but only TRIB1 and 2 can degrade it via the proteasome [4]. In lung cancer, TRIB2 blocks C/EBP-α activity by direct binding promoting the ubiquitination of C/EBP-α and consequently its proteasomal degradation through the interaction between the E3 ligase TRIM21 and the C-tail of TRIB2 [47]. Conversely, in AML another E3 ligase, COP1 is necessary for C/EBP-α degradation and the presence of the COP1 binding site in TRIB2 as well as the pseudokinase domain are essential to its oncogenic role in this caner type [14]. On the other hand, TRIB2 also modulates proteasome function via the proteasome subunit PSMB5 for which the PCBP2 protein is required. The DQLVPD sequence of TRIB2 binds to the KH3 domain of PCBP2 and PCBP2 interact with PSMB5, leading to the activation of PSMB5 and reducing Ub levels [48].

TRIB2 also promotes AKT activation via direct protein-protein interaction through the COP1 domain at the C-terminus of the protein. TRIB2 preferentially binds to catalytically inactive, non-phosphorylated threonine 308 AKT1 in vitro and increases endogenous AKT phosphorylation at the hydrophobic motif (serine 473) in human cancer cells [28]. TRIB2/AKT interaction promotes the activating phosphorylation on serine 473 of AKT without affecting its threonine 308 phosphorylation status. TRIB2-mediated AKT activation results in FOXO inhibition and increased MDM2 activity to degrade p53 [49]. As a consequence, the expression of FOXO and p53 target genes such as BIM, FasL and p21 which lead to drug induced apoptosis, is attenuated by TRIB2 [49,50,51].

## 4. TRIB2 in Health and Disease

The tribbles gene was first identified in *Drosophila* as a regulator of string⁄cdc25 in morphogenesis [16]. In Xenopus, Xtrb2 the TRIB2 orthologue was injected into embryos producing dorsal mesoderm involution defects, a phenotype that resembles gastrulation invagination defects seen in Drosophila trbl mutants [16]. In mice, TRIB2 expression is observed during gestation in different organs such as kidneys, mesonephros, testes, heart, eyes, thymus, blood vessels, muscle, bones, tongue, spinal cord, and ganglions. However, TRIB2 mutant mice show no apparent phenotypic variations, suggesting that it is not essential for murine development or that the other Tribbles proteins can compensate for the lack of TRIB2 [52].

Furthermore, the role of TRIB2 in stem cell differentiation has also been demonstrated. The siRNA-mediated suppression of TRIB2 in mesenchymal stem cells (MSCs) produced an increase in adipogenic differentiation by increasing C/EBP-α degradation [53,54]. Related to the role of TRIB2 in adipocytes, studies by Nakayama et al. in the Japanese population reported an adaptive variant of TRIB2, rs1057001, associated with high levels of thermogenic gene expression in both visceral and subcutaneous adipose tissue in humans [22]. This TRIB2 variant is implicated in visceral fat accumulation and is interpreted as the result of a positive natural selection of TRIB2 in East Asians [55].

The TRIB family has also been associated with reproduction. In cumulus cells that are surrounding bovine and mouse oocyte the expression levels of TRIB2 are the highest among the TRIBs. During oocyte maturation, TRIB2 expression levels are reduced and TRIB1 and TRIB3 are increased. This may be due to the different role of the TRIBs, which in the case of TRIB2 is more involved in proliferation and therefore is not necessary for oocyte maturation [56]. On the other hand, in granulosa cells, TRIB2 regulates gene expression, protein degradation and activation of signaling pathways that may contribute to follicular growth, differentiation of steroidogenic cells into luteal cells after ovulation and function of the corpus luteum [57].

At the level of hematopoiesis TRIB2 plays an important role mainly through the degradation of C/EBP-α. It has been observed that TRIB2 is overexpressed during thymus development, and in B cells, natural killers, and CD4+ T cells and to a lesser degree in CD8+ T cells [19]. Additionally, TRIB2 is expressed in early hematopoietic progenitors and erythroid precursors, and the deletion of TRIB2 mainly affected erythroid lineage development. TRIB2 has also been shown to promote erythropoiesis independent of C/EBPα proteins in vivo. As a result, TRIB2 knockout mice exhibit macrocytic anemia [58].

Consistent with the important role of TRIB2 in normal physiology, its deregulation can lead to disease. In inflammation TRIB2 has been described as a regulator of NF-κB activity associated with inflammatory bowel disease (IBD), and TRIB2 inhibits TLR5-mediated activation of NF-κB [29]. In fact, TRIB2 is significantly decreased in samples obtained from active IBD compared with inactive disease [29,59]. On the other hand, inflammatory activation of monocytes is an essential part of both innate immune responses and the pathogenesis of conditions such as atherosclerosis. The activation of monocytes measured by the production of IL-8 chemokine is MAPK dependent and downregulated by TRIB2 because of TRIB2 is a negative regulator of MAPK [60]. It should be noted that while modified low-density lipoprotein downregulates the expression of TRIB2 in monocytes [60], TRIB2 expression is upregulated by treatment with oxidized LDL in primary human monocyte-derived macrophages reducing IL-10. Considering that TRIB2 is a highly regulated gene in vulnerable atherosclerotic lesion, and that the inhibition of macrophage IL-10 biosynthesis is a potential pro-inflammatory consequence of high TRIB2 expression, TRIB2 might be associated to both inflammation and atherosclerosis [61].

## 5. TRIB2 in Cancer

The role of TRIB2 protein in cancer is currently emerging as a result of our growing understanding of its function in physiological processes and due to its diverse interactome. As summarized in Table 2, a multifunctional role of TRIB2 as an oncogenic protein has been described in numerous cancers where it is overexpressed, including melanoma [62], leukemias [63,64], pancreatic cancer [42], liver cancer [10], epithelial ovarian cancer [65] and lung cancer [47,66]. In addition, there is solid evidence that TRIB2 expression affects the sensitivity of cancer cells to chemotherapeutic and targeted anticancer drugs [49,67,68,69,70,71]. The oncogenic activity of TRIB2 is linked to its dysregulated expression, rather than to mutational activation, as in the case of melanoma and colorectal cancer, where TRIB2 is overexpressed and inversely correlated with survival of patients [49]. A recent study, however, reports a novel fusion transcript derived from a chromosomal translocation between the TRIB2 and the PRKCE genes, the latter coding for protein kinase C epsilon in pulmonary carcinoid tumours [66]. In cellular models of colorectal cancer (CRC), TRIB2 blocks cellular senescence by promoting transcriptional activity of AP4 and p21 repression [46]. However, the biological role of TRIB2 in CRC has not been elucidated, while TRIB2-mediated effects in leukemia, melanoma and liver cancer have been characterized more thoroughly.

### 5.1. TRIB2 in Leukemia

In the hematopoietic system, TRIB2 expression is important for lymphoid and erythroid lineages, as it is highly expressed in lymphoid populations, including B, T and NK cells, and moderately expressed in common myeloid progenitor (CMP) and in megakaryocyte–erythroid progenitors (MEP). Conversely, TRIB1 and TRIB3, are more abundant in myeloid lineage and primitive stem cells, respectively [19]. Although TRIB2 function is fundamental in lymphoid lineage, it is associated with both acute myeloid (AML) and acute lymphoblastic leukemias (ALL). TRIB2 is highly expressed in human T-ALL [63,64] where its expression correlates with Notch1 mutations in pediatric cases [63]. Additionally, TRIB2 is required for the growth and survival of human T-ALL cell lines driven by the oncogenic transcription factor TAL1 [33]. In contrast, Stein and colleagues found that knockdown of TRIB2 in murine T-ALL cell lines did not affect cell growth or survival and rather acts as tumour suppressor in Notch-driven T-ALL [77]. This finding suggests a more complex relationship between TRIB2 expression and T-ALL. TRIB2 is also potentially involved in B cell-ALL (B-ALL) with translocation t (1;19), as the expression level of TRIB2 in this subset of ALL was higher than that in T-ALL [63].

In the context of AML, TRIB2 was also shown to cooperate with HOXA9 (Homeobox A9) and accelerate murine AML [78]. HOXA9 and its cofactor Meis Homeobox 1 (MEIS1) are transcription factors expressed in early myeloid progenitors, which appear upregulated in AML [79]. Overexpression of Meis1 synergizes with multiple Nucleoporin 98 (NUP98) and HOX fusion genes to accelerate the onset of AML in murine bone marrow transplantation models [80,81]. TRIB2 is a target gene of HOXA9-mediated leukemogenesis and is activated by MEIS1 in AML driven by NUP98-HOXD13/MEIS1 rearrangements [35]. Additionally, several studies have shown that exogenous expression of TRIB2 confers a proliferative advantage in vitro and is sufficient to induce AML in vivo [82,83]. The best understood pathway by which TRIB2 promotes oncogenesis is its role in the degradation of C/EBP transcription factors [14,82,84]. Indeed, in the absence of C/EBPα, TRIB2 is unable to generate AML disease [83]. These effects are consonant with the association of inactivation of C/EBPα expression in human AML and the block in myeloid differentiation with the subsequently accumulation of myeloid blasts in the bone marrow observed in conditional mouse knockout models [85]. Moreover, TRIB2-dependent p42 isoform of C/EBPα degradation is mediated by the proteasome and constitutive photomorphogenesis 1 (COP1). Thus, COP1 is recruited by TRIB2 for the interaction with C/EBPα, suggesting TRIB2 functions as a scaffold protein to colocalize the enzyme with its target [14,86]. Nevertheless, Gilby and colleagues found that TRIB2 can act as tumour suppressor in AML and accordingly its expression was downregulated in patient samples [15]. Furthermore, TRIB2 seems not to be necessary for the initiation of myeloid leukemia, but is required for p38 MAPK signalling, a pathway that is activated in cellular stress conditions and regulates cell proliferation [45]. This tumour suppressive role of the TRIB2-p38 axis is also supported by the evidence that TRIB2 deficiency accelerates the onset of acute lymphoid leukemia [76].

The apparent contradiction that exists in the functionality of TRIB2 as an oncogene or tumour suppressor in AML models in vitro, might be due to the fact that both, high and low TRIB2 expression levels correlate with different types of leukemia in patients: AML subtypes with deletions in Chromosomes 5 or 7 display TRIB2 higher expression, whereas in AML subtypes with translocations of Chromosomes 9 or 11, TRIB2 expression is lower than in normal hematopoietic progenitor cells [19].

### 5.2. TRIB2 in Liver Cancer

Liver cancer is the sixth most common malignancy and the fourth most common cause of cancer-related mortality worldwide [87]. It is characterized by the dysregulation of metabolic and cell differentiation pathways including Wnt, ERK and AKT/mTOR signalling [88,89]. In the hepatocellular carcinoma context, TRIB2 has been shown to be a downstream effector and upstream regulator of Wnt signalling. TRIB2 is specifically activated by Wnt pathway effectors TCF4 and β-Catenin in the human liver cancer cell line HepG2 [38]. By contrast, TRIB2 facilitates the nuclear accumulation of its associated E3 ligases SMURF1, βTrCP and COP1 which leads with the ubiquitination and degradation of TCF4 and β-Catenin [12]. These findings suggest a negative auto-regulatory feedback in TRIB2-Wnt signalling in which TRIB2 acts as an oncogenic and tumour suppressive factor during liver tumorigenesis. Furthermore, recent studies have revealed a novel role of TRIB2 in modulating proteasome function by interaction with poly (rC)-binding protein 2 (PCBP2) and proteasome 20S subunit beta 5 (PSMB5). PCBP2 mediates degradation of some proteins and inhibits the production of reactive oxygen species (ROS), being implicated in development and progression of cancer. Mechanistically, TRIB2-PCBP2 interaction leads to a decrease in the ubiquitin pool by increasing the proteolytic efficiency of the proteasome and, consequently, maintains the viability of the liver cancer cells and promotes tumour growth [48].

Together, these data reveal the complexity of TRIB2 and ubiquitin proteasome system interactions in liver cancer cells and highlight the importance of cell context in TRIB2 protein function. Recent work in liver fibrosis and hepatocellular carcinoma (HCC) has demonstrated that TRIB2 is strongly upregulated in human fibrotic liver tissues and HCC tissues. TRIB2 colocalized with α-smooth muscle actin (α-SMA) in fibrotic and HCC liver tissues. Knockdown of TRIB2 inhibited hepatic stellate cells activation and liver fibrosis in vitro and in vivo [75], suggesting TRIB2 as an attractive therapeutic target for hepatic fibrosis and fibrosis-associated liver cancer.

### 5.3. TRIB2 in Melanoma

Melanoma is the tissue with the highest expression of TRIB2 and its expression level correlates with each melanoma stage [62]. In both, patient samples and ex vivo models, TRIB2 expression correlates also with treatment response, suggesting that the measurement of TRIB2 levels could be useful to both diagnose the disease and its grade. Moreover, TRIB2 has been identified as a repressor of FOXO proteins contributing to the growth and survival of melanoma cells [50]. In vivo, TRIB2 knockdown in melanoma xenograft models shows significant reduction of tumour growth [50]. In line with this, TRIB2 could be used as a biomarker in melanoma since its expression strongly correlates specifically with both the presence and progression of melanocyte-derived malignancies [62] but not in samples from other tumour sources, including non-basal skin carcinoma [50]. Recent work has revealed that TRIB2 positively correlates with the circular RNA 0084043 (circRNA_0084043). CircRNA is deregulated in melanoma tissues and cells and has an oncogenic role in human melanoma [72]. Apparently, circ_0084043 knockdown inhibited expression of β-catenin, c-Myc and cyclinD1 through downregulating TRIB2 via miR-429. Moreover, knockdown of circ_0084043 or TRIB2 suppressed melanoma development, indicating that they could be a potential therapeutic target [73].

Taken together, these data emphasize that TRIB2 plays a crucial role in regulating various cellular processes in cancer, such as proliferation, apoptosis and drug resistance [19,65,74] and although, currently, the role of TRIB2 in cancer remains controversial, we are moving forward in unraveling TRIB2 function in neoplasia.

## 6. The Emerging Role of TRIB2 in Therapy Resistance

In all areas of cancer therapy, resistance is a major concern, and it is broadly classified into two groups: acquired and intrinsic. Acquired drug resistance is a consequence of molecular changes and clonal selection observed after an extended period of drug treatment, whereas innate resistance occurs due to mutations in tumour cells during disease progression [90]. Table 3 outline the role of TRIB2 in cancer relapse due to chemotherapy resistance.

In several cancer cell lines, such as osteosarcoma and melanoma, as well as in vivo models and patient samples, TRIB2 overexpression has been shown to mediate resistance against various chemotherapeutics and targeted drugs including PI3K- and mTORC1-inhibitors, dacarbazine, gemcitabine and 5-fluorouracil, through the disruption of the AKT/FOXO and AKT/MDM2/p53 pathways. The increase of TRIB2 correlates with increased phosphorylation of AKT and MDM2 in patient samples and leads to a worse clinical outcome. Mechanistically, TRIB2 interacts via its COP1 domain with AKT [49]. TRIB2 preferentially binds to catalytically inactive, non-phosphorylated threonine 308 AKT1 in vitro and increases endogenous AKT phosphorylation at the hydrophobic motif (serine 473) in human cells [28]. It remains to be established whether TRIB2 indiscriminately binds to the AKT isoforms AKT1, AKT2 and AKT3 or with selective or preferential affinity. The phosphorylation of AKT results in an activation of its enzymatic activity and the subsequent inactivation of FOXO and p53 [49]. Thus, patients whose tumours display high TRIB2 expression may not benefit from specific PI3K inhibitor therapeutics, rendering TRIB2 as a suitable biomarker predicting treatment outcome and selecting patients for individualized therapy.

Furthermore, several recent articles suggest the implication of TRIB2 in the tumour sensitivity to cisplatin. Platinum-based drugs such as cisplatin are being used for the treatment of numerous cancer types including testicular, lung, ovarian, bladder, and head and neck cancers [91]. These drugs are classified as alkylating-like agents and act though damaging DNA [92]. Unfortunately, the great majority of cancer patients treated with platinum-containing drugs have some level of intrinsic resistance or develop acquired resistance to the treatment [93]. In cellular and in vivo models of small cell lung cancer, the cisplatin-resistant cells showed significantly higher expression of TRIB2 mRNA than control cells [67]. Moreover, the TRIB2 knockdown reverts cisplatin resistance, and correlates with upregulation of C/EBPα protein levels, indicating that TRIB2 could be a potential target to circumvent chemoresistance. Conversely, in epithelial ovarian cancer, TRIB2 knockdown increases resistance to cisplatin [65]. If, instead, the cells were treated with the hypomethylating agent 5′-Aza-Cytidine, TRIB2 was overexpressed in resistant cells [65]. One possible explanation for the TRIB2 knockdown-dependent cisplatin resistance may be due to the subsequent activation of FOXO proteins, since it has been reported that the levels of FOXO proteins are related to cisplatin resistance in ovarian cancer [94].

One of the anticancer drugs widely used in chemotherapy is doxorubicin (also known as adryamicin). It is a 14-hydroxylated analogue of daunorubicin, a natural product also used as anticancer drug [95]. These molecules act by intercalating in the DNA causing breakage of DNA strands and inhibition of DNA synthesis [96]. In AML, Ma and colleagues have demonstrated that in doxorubicin resistant cells, TRIB2 knockdown decreases IC50 and increases intracellular doxorubicin accumulation [68]. Additionally, they demonstrate that TRIB2 promotes cellular efflux functions of doxorubicin by increasing the MDR1 and MRP1 cell membrane transporters, considered as “drug resistance proteins” because they work as drug efflux pump mediating multidrug resistance of human leukemia cells [97]. Consistent with this observation, a recent study uncovered an additional mechanism of chemoresistance in AML in which TRIB2 overexpression confers resistance to cytarabine, doxorubicin and daunorubicin, DNA intercalating agents which cause breakage of DNA strands and inhibition of DNA synthesis [70]. Nevertheless, as mentioned before, TRIB2 has been shown to act both as an oncogenic driver and tumour suppressor in AML [19,82,83]. Thus, TRIB2 deficiency conferred resistance to daunorubicin treatment by promoting growth and survival advantage in myeloid leukemia cells [45]. TRIB2 re-expression or pharmacological activation of p38 MAPK signalling in TRIB2 deficient leukemia cells sensitized the cells to chemotherapy-induced apoptosis.

The wealth of conflicting studies highlights the need and importance of carefully approaching the analysis of TRIB2 function, as it exerts a dual role in myeloid leukemia and liver in a context-specific manner. It should also be noted that Tribbles proteins can perform redundant functions, which may mask some of the effects seen in these studies. For example, TRIB1 also accelerates degradation of C/EBPα in AML cells [98]. In addition, TRIB1 is also involved in the etiology of glioma [99], liver [100] and prostate cancer [101]. Regarding TRIB3, its expression was found elevated in patients with colorectal cancer and its expression correlated with poor overall survival [102].

## 7. Targeting TRIB2

Due to the importance of TRIB2 in disease progression and therapeutic resistance, TRIB2 presents an exciting opportunity for anti-cancer drug development efforts. TRIB2 is an unexplored, druggable therapeutic target at the core of a clinically relevant drug resistance mechanisms. Bailey et al. reported that TRIB2 binds to ATP and autophosphorylates suggesting that small molecule kinase inhibitors might be capable of manipulating TRIB2 function [13]. According to the specific oncogenic context, both up- or down-regulation of TRIB2, are the two approaches for targeting TRIB2. Therefore, so far, strategies to target TRIB2 are based on pharmacological approaches manipulating the endogenous levels of TRIB2 indirectly by modifying its direct regulators including miRNAs and proteins that regulate TRIB2 degradation. Importantly, TRIB2 is a target of small-molecule protein kinase inhibitors, originally designed to interfere with HER2 and EGFR tyrosine kinase family members [28]. Second and third generation dual HER2 and EGFR inhibitors including afatinib, neratinib and osimertinib (known to irreversibly inhibit these receptor tyrosine kinases by covalently binding to a cysteine residue) also interact with a cysteine residue present in the kinase domain of TRIB2 in a covalent manner [28]. Unlike related noncovalent inhibitors like lapatinib, these covalent inhibitors lead to rapid degradation of the TRIB2 protein [28]. Moreover, this inhibition is specific for TRIB2 because of the lack of equivalent Cys residues in other pseudokinases, including TRIB1, TRIB3, and STK40, indicating the convenience of this therapeutic approach for tumours in which high levels of TRIB2 correlates with worst prognosis. It is worth noting that these covalent inhibitors have been approved for their clinical use to treat patients with non-small cell lung cancer or breast cancer. Further to this, using RNA sequencing-based transcriptional profiling of isogenic osteosarcoma U2OS cells with different TRIB2 status, a recent study identified several small-molecule compounds, including harmine and piperlongumine, that reverse TRIB2-dependent transcriptional signatures [71]. Consistent with the involvement of TRIB2 in promoting PI3K/AKT signalling, these two naturally occurring alkaloids induce the nuclear translocation of the FOXO3 transcription factor. As TRIB2 expression level correlates with increased resistance to chemotherapy, this work suggests a therapeutic combination of harmine or piperlongumine with the dual PI3K/mTOR inhibitor BEZ235 treatment to overcome resistance by increasing the cytostatic effects of BEZ235. Targeting TRIB2 could also be achieved by using miRNAs. It has been reported that miR-206 and miR-140 act as tumour suppressor in lung adenocarcinoma by modifying oncogenic TRIB2 promoter activity [34]. This study shows that miR-206 and miR-140 target the transcription factor phospho-Smad3 which, in turn, binds CAGACA sequences in TRIB2 promoter. In line with this data, in osteosarcoma tissues and cell lines, TRIB2 is a direct target of miR-509-5p and when miR-509-5p is overexpressed or TRIB2 is silenced, the malignant capacity of osteosarcoma cells, in terms of proliferation and invasion, its reduced [103]. Additionally, TRIB2 stability could be target by modifying Ubiquitin E3 ligase SCF^β-TRCP^ levels. SCF^β-TRCP^ is a multiprotein complex composed by the F-box protein β-TRCP, scaffold protein CUL1 and the adapter protein SKP1. This complex polyubiquitinate TRIB2 on its degradation domain (TDD) located at the N-terminal domain, triggering its proteasomal degradation [27]. Targeted protein degradation using proteolysis targeting chimaeras (PROTACs) [104] or hydrophobic tagging (HyT) [105] that promote the degradation of specific proteins has attracted extensive interest as a modality for modulating ‘undruggable’ targets. In order to apply these approaches to enable targeting of TRIB2, selective and potent ligands need to be identified as starting point for PROTAC/HyT discovery.

## 8. TRIB2 as a Biomarker

Early diagnosis of many pathologies, such as cancer, can make a difference in the effectiveness of treatment and the prognosis of a disease. It has been largely demonstrated that several proteins are significantly upregulated in a disease-dependent manner. The proteins that could be taken as signatures for diagnostic confirmation are known as biomarkers. The above-described differential expression of TRIB2 in several cancer types and its association with drug resistance, suggests TRIB2 as a powerful biomarker. TRIB2 exhibits low expression in healthy skin samples, increases in benign melanoma, with the highest expression seen in malignant melanoma samples [62]. Given the correlation between TRIB2 expression levels and the stage of melanoma, TRIB2 could be used as a biomarker for diagnosis and progression. Moreover, TRIB2 overexpression implies tumour resistance to PI3K/mTOR inhibitors, so it might also be useful as a predictive biomarker in a personalized therapy context. TRIB2 had also showed its potential as a combined biomarker in concurrence with other proteins, such as BCL2 in AML or MAP3K in glioblastoma. In AML, TRIB2 overexpression results in an increase in BCL2 expression and these cells can be target by BCL2 inhibition with venetoclax (ABT-199), a BCL2-domain mimetic [70]. These results suggest that, using TRIB2 as biomarker, combination treatment with conventional chemotherapy (daunorubicin) and BCL2 inhibition would overcome the drug resistance. Moreover, recent work in glioblastoma has revealed, not only that high levels of TRIB2 correlates with poor prognosis of patients with glioblastoma, but also that combined increased in TRIB2 and MAP3K is associated with resistance to temozolomide (TMZ) and radiotherapy [69], which are the standard of care for this tumour type. Combined elevation of TRIB2 and MAP3K1 could be used as novel prognostic biomarkers to evaluate the malignancy and long-term outcomes of glioblastoma. Thus, use of TRIB2 as biomarker has the potential to stratify clinical approach by identifying patients who may benefit most from specific lines of treatment.

## 9. Conclusions

The early detection of a disease as well as the development of an effective and personalized therapy arises as a consequence of the detailed knowledge of key proteins in these pathological processes. Therefore, in this review, we wanted to highlight that TRIB pesudokinases present a new and potential pharmaceutical and therapeutic opportunity in various pathologies, but especially in cancer. We focus on TRIB2 because, even though its crystallographic structure is not available, a lot of work is being done in unraveling its functions. The highlight of TRIB2 structure is its pseudokinase domain, which has been evolutionary conserved and is homologous to serine/threonine kinase domains but lacks the lysine active site. TRIB2 pseudokinase domain, as in canonical kinases, is formed by two lobes, the N-lobe and the C-lobe. The N-lobe is involved in TRIB2 homeostasis and the C-lobe is the substrate-binding site. The most C-terminal region of TRIB2 is the best described part of the protein. This domain stands out for containing TRIB2 protein-protein interaction sequences with E3-ligases. Thus, TRIB2 acts as a scaffold of the ubiquitin-proteasome system for protein degradation.

It should also be noted, that a great variety of TRIB2 interactors have been described in a cell context-dependent manner. For example, while TRIB2 interacts with NF-κB in epithelial cells, it interacts with OCT4 in embryonic stem cells, although the exact structural region of TRIB2 responsible for this interaction is still elusive. In addition to this, we have also summarized the multiple interactors of TRIB2 in a disease context. In colorectal cancer, TRIB2 interacts with AP4 protein through the pseudokinase domain. Similarly, TRIB2 also activates the p38 MAPK pathway in myeloid leukemia. Moreover, TRIB2 levels appear to correlate with many neoplasia conditions, conferring TRIB2 a role as a biomarker, which in contexts such as melanoma is even dependent on cancer stage progression and prognosis. It is important to note that the cause of TRIB2 overexpression in neoplasias has not been elucidated, although we know that TRIB2 is strictly regulated at both the transcriptional level and protein level. As we have also detailed in this review, TRIB2 levels are regulated through miRNAs, as well as by the phosphorylation of its serine 83 by p70S6K. Furthermore, TRIB2 degradation is also carried out by SMURF1 and βTRCP. In order to understand differential expression and activity of TRIB2 in cancer, the network of components acting upstream of TRIB2 have to be dissected and carefully examined in vitro and in vivo. Similarly, the use of mouse models including murine xenograft and immunocompetent allograft models as well as inducible, tissue specific deletion and overexpression of TRIB2 will help to further define the specific role of TRIB2 in the formation, progression and therapy resistance of different tumour types. The multilayer regulation of TRIB2 homeostasis provides the opportunity to target TRIB2, specially in cases where TRIB2 expression confers resistance to therapy.

## Figures and Tables

**Figure 1 cancers-13-02701-f001:**
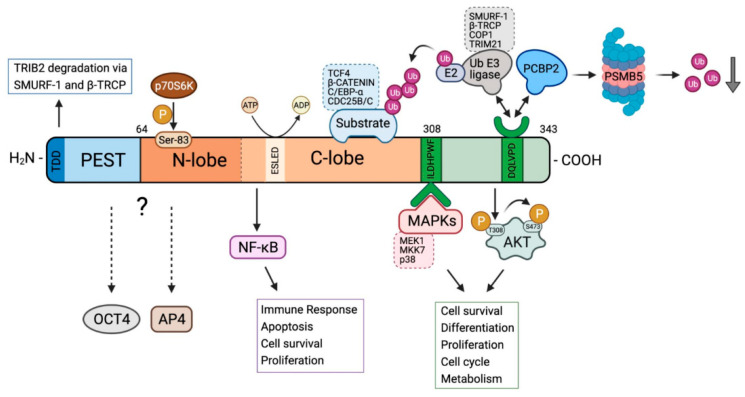
Structure and function of TRIB2 pseudokinase. TRIB2 is a three-domain protein containing an N-terminal PEST region (blue), a bilobed pseudokinase domain (orange) and a C-terminal tail (green). The PEST sequence and the five most N-terminal amino acids (TRIB2 Degradation Domain, TDD) are involved in TRIB2 degradation. The N-lobe of pseudokinase domain is characterized by the phosphorylation of serine 83 by p70S6K, which is also involved in TRIB2 homeostasis. The C-lobe of pseudokinase domain contains the ESLED sequence which is implicated in ATP binding and hydrolysis. The interaction of TRIB2 with NF-κB and with substrates of the ubiquitin-proteasome system (TCF4, β-CATENIN, C/EBPα and CDC25B/C) occurs also through C-lobe of pseudokinase domain. Within the C-terminal region of TRIB2, there are two important conserved sequences: ILDHPWF sequence is involved in the interaction with MEK1, MKK7 and p38 MAPKs, and the sequence DQLVPD is related to the interaction with Ubiquitin E3-ligases such as SMURF-1, βTRCP, COP1 and TRIM21, that polyubiquitylate the substrates attached to pseudokinase domain. The DQLVPD sequence also binds PCBP2 protein, which interacts with PSMB5 proteasome subunit leading to a reduction in global ubiquitin levels. The C-terminal domain is also involved in the direct interaction with non-phosphorylated threonine 308 of AKT, which promotes the phosphorylation of serine 473 of AKT and enhances its activation. The dashed arrows indicate that it has been described that OCT4 and AP4 interact with TRIB2 but is has not been clarified with which region of the protein.

**Table 1 cancers-13-02701-t001:** Summary of transcription factors demonstrated to regulate TRIB2.

Transcription Factor	Regulation	Cell Type	References
E2F1	Positive	Acute myeloid leukemia (AML) mice	[32]
C/EBPα-p30	Positive	Acute myeloid leukemia (AML) mice	[32]
TAL1	Positive	Acute lymphoblastic leukemia (ALL) human	[33]
NOTCH1	Positive	Acute lymphoblastic leukemia (ALL) human	[33]
Smad3	Positive	Lung adenocarcinoma human	[34]
Mesi1	Positive	Acute myeloid leukemia (AML) mice	[35]
PITX1	Positive	Acute lymphoblastic leukemia (ALL) human	[36]
FOG-1	Positive	Bone marrow mice	[37]
GATA-1	Positive	Bone marrow mice	[37]
TCF	Positive	Liver cancer cells human	[38]
FoxA	Positive	Liver cancer cells human	[38]
C/EBPα-p42	Negative	Acute myeloid leukemia (AML) mice	[32]
E2A	Negative	Acute lymphoblastic leukemia (ALL) human	[33]

**Table 2 cancers-13-02701-t002:** Summary of TRIB2 function in different human cancers.

Cancer Type	TRIB2 Function	Target	Cell Line/Model	Mode of Action	References
Lung	Oncogenic	C/EBPα, pSmad3, TRIM21	A549, LTEP-a-2 cells NOD/SCID mice Patients’ tissue samples	Promotes proliferation, sphere formation and in vivo tumorigenicity	[34,47,66]
Colorectal	Oncogenic	AP4/p21, AK	SW48 and LoVo cells Primary tumour samples	Blocks cellular senescence Confers resistance to PI3K inhibitors	[46,49]
Pancreatic	Oncogenic	miR505/ ZEB1-AS1	SW1990 and Capan-1 cells	Promotes viability, migration and invasion	[42]
Ovarian	Tumour suppressor	p21	A2780 cell line Patient tumour samples	Induces of a cisplatin-dependent cell cycle arrest and apoptosis	[65]
Melanoma	Oncogenic	FOXO, β-catenin, c-Myc and cyclinD1	Sk-Mel-28, Sk-Mel-94, Sk-Mel-19, UACC-257, Hs 895T, Sk-Mel-147, A375 and HaCaT cells Melanoma xenograft	Promotes proliferation and drug resistance. Inhibition of apoptosis	[50,72,73,74]
Liver	Oncogenic	βTrCP, C/EBPα, SMURF1, PCBP2	HepG2, Bel-7402, Bel-7404, Hep3B, Huh7, SMMC-7721, Bel-7402 and HL-7702 cells Mouse model of liver fibrosis and HCC	Promotes tumorigenesis, tumour growth and cell survival	[10,12,48,75]
Leukemia	Oncogenic	C/EBPα	32D cells C/EBPα^fl/fl^ mouse model	Cooperates with HOXA9 in AML Confers proliferative advantage and induce AML	[63,64]
	Tumour suppressor	Notch p38	Patient samples	Is required for p38 MAPK signalling, induction of cell cycle checkpoint response and apoptosis. Its deficiency accelerates the onset of ALL	[15,45,76]

**Table 3 cancers-13-02701-t003:** Role of TRIB2 in chemotherapy resistance.

Tumour Type	Anticancer Drug	Cell Line/Model	Role of TRIB2 in Resistance	References
Melanoma Osteosarcoma	PI3K inhibitors mTOR inhibitors Dacarbazine Gemcitabine 5-fluorouracil	U2OS osteosarcoma cell line Patient melanoma samples	TRIB2 disrupts AKT/FOXO signalling by increasing AKT and MDM2 phosphorylation	[49]
Glioblastoma	Temozolomide	Patient tumour samples	Combined increase in TRIB2 and MAP3K1	[69]
Small Cell Lung Cancer	Cisplatin	NCI-H69 xenograft mouse model, Patient tumour samples	TRIB2 overexpression increases C/EBPα downregulation	[67]
Epithelial Ovarian Cancer	Cisplatin	A2780 cell line, Patient tumour samples	There is a correlation with TRIB2 knockdown and resistance	[65]
	5′-Aza-Cytidine	A2780 Cisplatin resistant cell line	TRIB2 is overexpressed in resistant cells	
Acute Myeloid Leukemia	Doxorubicin	K562 cell line	TRIB2 promotes the expression of cell membrane transporters MRP1 and MDR1 increasing the efflux of drug	[68]
	Cytarabine Daunorubicin	K562 cell line	TRIB2 overexpression increases BCL2 expression, promoting antiapoptic signalling	

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
