# Peer review of "The Critical Role of TRIB2 in Cancer and Therapy Resistance"

_cancers, 2021, doi:10.3390/cancers13112701_

Round 1

Reviewer 1 Report

The manuscript by Mayoral-Varo et al provides a thorough and up-to-date summary of the involvement of TRIB2 in cancer biology. The manuscript is well-written and deserves praise for covering different aspects of TRIB2 biology, including protein structure, gene regulation and cell biology. Particularly in the recent years, there has been an accumulation of TRIB2 literature, warranting a summarization such as this manuscript.

As with any emerging protein-of-interest, there are apparently contradictory results regarding some TRIB2 functions (such as role in leukemia) and much evidence is at the level of monolayer cell cultures (Table 2). The authors could add a couple on sentences (for instance, to the Conclusions section) stating which model systems need to be explored next in the TRIB2 field in order to further increase the value of TRIB2 as a cancer treatment target. For instance, most cancer types in Table 2 appear currently to lack xenograft evidence – is this something that awaits to be done? What about tumor models where the host immune system is functional, unlike classical xenografts? From the in vitro models, would 3D cell cultures (organoids) and co-cultures be warranted?

The text mentions “AKT” but sometimes also specifically “AKT1”. Is there any data regarding TRIB2 interaction with AKT proteins other than AKT1? If interactions with AKT2 and AKT3 have been studied, it could be worth mentioning whether the results were matching to AKT1 results or not. If this is unknown, should it be studied in the future?

Line 64: “… but lack the lysine active site [7].“ – please double-check this claim and specify, for instance, the species that is meant (mammals or Drosophila), or, please specify which Lys residue is meant. For instance, Hegedus et al (2007; Ref 11) in their Figure 4 present a protein sequence alignment of various TRIB family proteins. In the alignment by Hegedus et al (2007), subdomain II of the kinase domain of Drosophila Tribbles indeed lacks the Lys residue (having there an Arg). However, most mammalian TRIB proteins have there the Lys residue.

Lines 263-267: The potential involvement of TRIB2 in anxiety and narcolepsy, while interesting, is perhaps too distant from the topic of the current review and could be removed. It appears that for the other TRIB2 roles described in section 4 (“TRIB2 in health and disease”), the cell type or tissue discussed is generally also one where TRIB2 has some involvement in cancer of that origin. Alternatively, could TRIB2 in anxiety and narcolepsy be related to the glioblastoma effect mentioned in Table 3?

Section 7: Could TRIB2 be a good candidate for targeting via PROTAC (Proteolysis- targeting chimaera), HyT (hydrophobic tagging) or related technologies?

Minor issues and suggestions:

Line 20: could be clearer to state “Tribbles proteins” or “TRIB proteins” instead of just “Tribbles” (which could be understood as referring to only the fruit fly gene or protein in singular form).

Line 22: Same as above: perhaps replace “Tribbles“ with “Tribbles proteins” or “TRIB proteins“ for clarity.

Line 42: “orthologs was identified“ should be „orthologs were identified“.

Line 50: „aminoacids“ missing space.

Line 60: „±25 residues“ would be clearer to state “approximately 25 residues“.

Line 74: “The conservated Lys 90 and 177 residues …“ should be “The conserved Lys 90 and 177 residues …“.

Line 104: „aminoacid“ missing space.

Line 119: “phosphorylated non-threonine 308 of AKT“  has some wording issue (“non-threonine”). Is it meant AKT1 unphosphorylated at Thr308?

Line 122: excess comma after “domain”.

Line 206: “non-threonine 308 phosphorylated” possibly needs an extra hyphen, giving “non-Thr308-phosphorylated“.

Line 215: “xenopus”: capitalize to “Xenopus”

Line 249: missing “s” in “TRIB2 inhibits“. Also, this sentence is missing a literature reference.

Line 251: since the abbreviation “IBD” was defined above, “inflammatory bowel disease” could be shortened here as “IBD”.

Line 256: excess comma after “while”.

Line 257: typo in “TIRB2”.

Line 267: “patients narcolepsy” should be “narcolepsy patients”.

Table 2: “Cooperate with HOXA9 in AML” should be “Cooperates…”

Table 2: part of sentence or word missing: “Confers proliferative advantage an induce AML“.

Line 312: missing “s”, should be “TRIB2 promotes”.

Line 313: should be “TRIB2 is” not “TRIB2 in”.

Line 364: “measure TRIB2” should be, for example “the measurement of TRIB2”.

Table 3: “TRIB2 disrupt” should be “TRIB2 in disrupts”

Table 3: perhaps it would be useful to add a “Cell line/Model” column like in Table 2. This would distinguish which ones have in vivo or in vitro level of evidence.

Line 397: hyphen missing; should be, for example, “non-Thr308-phosphorylated AKT1”.

Line 410: “small lung cancer“ should probably be „small cell lung cancer“.

Line 423: “strands an inhibition“ should probably be “strands and inhibition“.

Line 432: “strands an inhibition“ should probably be “strands and inhibition“.

Lines 458-461: the sentence needs extra punctuation for readability, for instance, adding parenthesis as such: “Second and third generation dual HER2 and EGFR inhibitors including  afatinib, neratinib and osimertinib (known to irreversibly  inhibit  these receptor  tyrosine  kinases by covalently binding to a cysteine residue) also interact with a cysteine residue  present in the kinase domain of TRIB2 in a covalent manner.” Additionally, this sentence is missing literature references.

Line 541: excess word: “that“

Reference 5 (lines 561-562): this reference appears to have the wrong year listed, and the journal name is missing.

Author Response

Dear Reviewer,

Thank you very much for your time and highly constructive comments that have no doubt strengthened our manuscript further. We have carefully considered comments and suggestions raised and we believe that by addressing all of the comments raised, that this manuscript would now be very suitable for publication in Cancers.

                        To assist you, the changes made to the manuscript are highlighted with track changes in this modified version and they are detailed below point by point. We have also included all of the reviewer comments that our original manuscript received (in italic) and our response to each comment (bold).

Reviewers' comments:

The manuscript by Mayoral-Varo et al provides a thorough and up-to-date summary of the involvement of TRIB2 in cancer biology. The manuscript is well-written and deserves praise for covering different aspects of TRIB2 biology, including protein structure, gene regulation and cell biology. Particularly in the recent years, there has been an accumulation of TRIB2 literature, warranting a summarization such as this manuscript.

Response

We thank the reviewer for the positive and very helpful feedback

 As with any emerging protein-of-interest, there are apparently contradictory results regarding some TRIB2 functions (such as role in leukemia) and much evidence is at the level of monolayer cell cultures (Table 2). The authors could add a couple on sentences (for instance, to the Conclusions section) stating which model systems need to be explored next in the TRIB2 field in order to further increase the value of TRIB2 as a cancer treatment target. For instance, most cancer types in Table 2 appear currently to lack xenograft evidence – is this something that awaits to be done? What about tumor models where the host immune system is functional, unlike classical xenografts? From the in vitro models, would 3D cell cultures (organoids) and co-cultures be warranted?

Response

This is an excellent point raised by this reviewer. In response, we added a short discussion of these topics to the conclusion section of the revised manuscript

The text mentions “AKT” but sometimes also specifically “AKT1”. Is there any data regarding TRIB2 interaction with AKT proteins other than AKT1? If interactions with AKT2 and AKT3 have been studied, it could be worth mentioning whether the results were matching to AKT1 results or not. If this is unknown, should it be studied in the future?

Response

This is a very relevant point. Currently, it is not clear whether TRIB2 indiscriminately binds to the AKT isoforms AKT1, AKT2 and AKT3 or with selective or preferential affinity. In the revised version of the manuscript, we mention this as a future challenge to be investigated.

 Line 64: “… but lack the lysine active site [7].“ – please double-check this claim and specify, for instance, the species that is meant (mammals or Drosophila), or, please specify which Lys residue is meant. For instance, Hegedus et al (2007; Ref 11) in their Figure 4 present a protein sequence alignment of various TRIB family proteins. In the alignment by Hegedus et al (2007), subdomain II of the kinase domain of Drosophila Tribbles indeed lacks the Lys residue (having there an Arg). However, most mammalian TRIB proteins have there the Lys residue.

Response

We are grateful to the reviewer for drawing our attention to this issue. In response, we removed this statement and added a sentence highlighting that the most striking difference between conventional kinases and the pseudokinase domain of Tribbles proteins is the lack of a DFG triplet.

 Lines 263-267: The potential involvement of TRIB2 in anxiety and narcolepsy, while interesting, is perhaps too distant from the topic of the current review and could be removed. It appears that for the other TRIB2 roles described in section 4 (“TRIB2 in health and disease”), the cell type or tissue discussed is generally also one where TRIB2 has some involvement in cancer of that origin. Alternatively, could TRIB2 in anxiety and narcolepsy be related to the glioblastoma effect mentioned in Table 3?

Response

We appreciate these important comments. In response, we removed the statements referring to the role of TRIB2 in anxiety and narcolepsy from the revised versions of the manuscript. The idea that there might be a connection between the involvement of TRIB2 in anxiety and narcolepsy with its expression in specific cancer types is intriguing, but might be too speculative in the context of this review.

 Section 7: Could TRIB2 be a good candidate for targeting via PROTAC (Proteolysis- targeting chimaera), HyT (hydrophobic tagging) or related technologies?

Response

This is an excellent suggestion. In response, we added statements and two references to the revised manuscript discussing these approaches in the context of TRIB2.

Minor issues and suggestions:

Line 20: could be clearer to state “Tribbles proteins” or “TRIB proteins” instead of just “Tribbles” (which could be understood as referring to only the fruit fly gene or protein in singular form).

Response

We agree with this comment and in response, we consistently use Tribbles proteins throughout the revised version of the manuscript.

Line 22: Same as above: perhaps replace “Tribbles“ with “Tribbles proteins” or “TRIB proteins“ for clarity.

Response

We agree with this comment and in response, we consistently use Tribbles proteins throughout the revised version of the manuscript.

Line 42: “orthologs was identified“ should be „orthologs were identified“.

Response
We thank the reviewer for drawing our attention to this mistake. We corrected this in the revised text.

Line 50: „aminoacids“ missing space.

Response
We thank the reviewer for drawing our attention to this mistake. We corrected this in the revised text.

Line 60: „±25 residues“ would be clearer to state “approximately 25 residues“.

Response
We followed the suggestion of the reviewer

Line 74: “The conservated Lys 90 and 177 residues …“ should be “The conserved Lys 90 and 177 residues …“.

Response
We thank the reviewer for drawing our attention to this mistake. We corrected this in the revised text.

Line 104: „aminoacid“ missing space.

Response
We thank the reviewer for drawing our attention to this mistake. We corrected this in the revised text.

Line 119: “phosphorylated non-threonine 308 of AKT“  has some wording issue (“non-threonine”). Is it meant AKT1 unphosphorylated at Thr308?

Response
We thank the reviewer for drawing our attention to this mistake. We corrected this in the revised text.

Line 122: excess comma after “domain”.

Response
We thank the reviewer for drawing our attention to this mistake. We corrected this in the revised text.

Line 206: “non-threonine 308 phosphorylated” possibly needs an extra hyphen, giving “non-Thr308-phosphorylated“.

Response
We thank the reviewer for drawing our attention to this mistake. We corrected this in the revised text.

Line 215: “xenopus”: capitalize to “Xenopus”

Response
We thank the reviewer for drawing our attention to this mistake. We corrected this in the revised text.

Line 249: missing “s” in “TRIB2 inhibits“. Also, this sentence is missing a literature reference.

Response
We thank the reviewer for drawing our attention to this mistake and the required reference. We corrected this in the revised text.

Line 251: since the abbreviation “IBD” was defined above, “inflammatory bowel disease” could be shortened here as “IBD”.

Response
We thank the reviewer for drawing our attention to this mistake. We corrected this in the revised text.

Line 256: excess comma after “while”.

Response
We thank the reviewer for drawing our attention to this mistake. We corrected this in the revised text.

Line 257: typo in “TIRB2”.

Response
We thank the reviewer for drawing our attention to this mistake. We corrected this in the revised text.

Line 267: “patients narcolepsy” should be “narcolepsy patients”.

Response

We removed the statements referring to the role of TRIB2 in anxiety and narcolepsy from the revised versions of the manuscript.

Table 2: “Cooperate with HOXA9 in AML” should be “Cooperates…”

Response
We thank the reviewer for drawing our attention to this mistake. We corrected this in the revised text.

Table 2: part of sentence or word missing: “Confers proliferative advantage an induce AML“.

Response
We thank the reviewer for drawing our attention to this mistake. We corrected this in the revised text.

Line 312: missing “s”, should be “TRIB2 promotes”.

Response
We thank the reviewer for drawing our attention to this mistake. We corrected this in the revised text.

Line 313: should be “TRIB2 is” not “TRIB2 in”.

Response
We thank the reviewer for drawing our attention to this mistake. We corrected this in the revised text.

Line 364: “measure TRIB2” should be, for example “the measurement of TRIB2”.

Response
We thank the reviewer for drawing our attention to this mistake. We corrected this in the revised text.

Table 3: “TRIB2 disrupt” should be “TRIB2 in disrupts”

Response
We thank the reviewer for drawing our attention to this mistake. We corrected this in the revised text.

Table 3: perhaps it would be useful to add a “Cell line/Model” column like in Table 2. This would distinguish which ones have in vivo or in vitro level of evidence.

Response

We thank the reviewer for this suggestion. We included the extra column in table 3 of the revised version of the manuscript

Line 397: hyphen missing; should be, for example, “non-Thr308-phosphorylated AKT1”.

Response
We thank the reviewer for drawing our attention to this mistake. We corrected this in the revised text.

Line 410: “small lung cancer“ should probably be „small cell lung cancer“.

Response
We thank the reviewer for drawing our attention to this mistake. We corrected this in the revised text.

Line 423: “strands an inhibition“ should probably be “strands and inhibition“.

Response
We thank the reviewer for drawing our attention to this mistake. We corrected this in the revised text.

Line 432: “strands an inhibition“ should probably be “strands and inhibition“.

Response
We thank the reviewer for drawing our attention to this mistake. We corrected this in the revised text.

Lines 458-461: the sentence needs extra punctuation for readability, for instance, adding parenthesis as such: “Second and third generation dual HER2 and EGFR inhibitors including  afatinib, neratinib and osimertinib (known to irreversibly  inhibit  these receptor  tyrosine  kinases by covalently binding to a cysteine residue) also interact with a cysteine residue  present in the kinase domain of TRIB2 in a covalent manner.” Additionally, this sentence is missing literature references.

Response

We followed this suggestion, modified the sentence and added the required reference.

Line 541: excess word: “that“

Response
We thank the reviewer for drawing our attention to this mistake. We corrected this in the revised text.

Reference 5 (lines 561-562): this reference appears to have the wrong year listed, and the journal name is missing.

Response

We thank the reviewer for drawing our attention to this inconsistency. We corrected the reference accordingly.

Once again, we would like to thank the reviewer for your time and effort

Sincerely,

Wolfgang Link

Reviewer 2 Report

A review paper by Link et al. presents and discusses the role of TRIB pseudokinases in different types of cells, especially cancer. The manuscript is very well written, it is very informative and perfectly structured. The Authors use clear English, and they refer to appropriate references. All pieces of information are presented concisely. In general, this is a very good review paper. I have only minor comments. 

Minor comments:

  1. Paragraph 3/title - please add "regulation of TRIB2 .... (expression/level?) and TRIB2 signaling".
  2. Please unify the way of preparing tables (upper vs lower letter case), e.g.  Table 3: Small lung cancer vs Acute Myeloid Leukemia.

Author Response

Dear Reviewer,

Thank you very much for your time and highly constructive comments that have no doubt strengthened our manuscript further. We have carefully considered comments and suggestions raised and we believe that by addressing all of the comments raised, that this manuscript would now be very suitable for publication in Cancers.

                        To assist you, the changes made to the manuscript are highlighted with track changes in this modified version and they are detailed below point by point. We have also included all of the reviewer comments that our original manuscript received (in italic) and our response to each comment (bold).

Reviewers' comments:

A review paper by Link et al. presents and discusses the role of TRIB pseudokinases in different types of cells, especially cancer. The manuscript is very well written, it is very informative and perfectly structured. The Authors use clear English, and they refer to appropriate references. All pieces of information are presented concisely. In general, this is a very good review paper. I have only minor comments. 

Response

We thank the reviewer for the positive and very helpful feedback

Minor comments:

Paragraph 3/title - please add "regulation of TRIB2 .... (expression/level?) and TRIB2 signaling".

Response

We thank the reviewer for this suggestion. We modified the title accordingly.

Please unify the way of preparing tables (upper vs lower letter case), e.g.  Table 3: Small lung cancer vs Acute Myeloid Leukemia.

Response
We thank the reviewer for drawing our attention to this inconsistency. We modified the letter case accordingly.

Once again, we would like to thank the reviewer for your time and effort

Sincerely,

Wolfgang Link
